# mBLIP: Efficient Bootstrapping of Multilingual Vision-LLMs

## Abstract

Modular vision-language models (Vision-LLMs) align pretrained image encoders with frozen large language models (LLMs), representing a computationally much more efficient alternative to end-to-end training of large vision-language models from scratch, which is prohibitively expensive for most researchers and practitioners. Vision-LLMs instead post-hoc condition LLMs to 'understand' the output of an image encoder. With the abundance of readily available high-quality English image-text data as well as monolingual English LLMs, the research focus has been on English-only Vision-LLMs. Multilingual vision-language models are still predominantly obtained via expensive end-to-end pretraining, resulting in comparatively smaller models, trained on limited multilingual image data supplemented with text-only multilingual corpora. In this work, we present mBLIP, the first multilingual Vision-LLM, which we obtain in a computationally efficient manner – on consumer hardware and using only a few million training examples – by leveraging a pretrained multilingual LLM. To this end, we *re-align* an image encoder previously tuned to an English LLM to a new, multilingual LLM – for this, we leverage multilingual data from a mix of vision-and-language tasks, which we obtain by machine-translating high-quality English data to 95 languages. On the IGLUE benchmark, mBLIP yields results competitive with state-of-the-art models. Moreover, in image captioning on XM3600, mBLIP (zero-shot) even outperforms PaLI-X (a model with 55B parameters). Compared to these very large multilingual vision-language models trained from scratch, we obtain mBLIP by training orders of magnitude fewer parameters on magnitudes less data. We release our model and code at `ANONYMIZED`.

## 1 Introduction

The success of model and data scaling in NLP from BERT (Devlin et al., 2019) to more recent Large Language Models (LLMs) (Brown et al., 2020; Zhang et al., 2022; Touvron et al., 2023, *inter alia*) has prompted similar endeavors in vision-language pretraining from 'small' BERT-size models (Chen et al., 2020; Li et al., 2020; 2021; 2022) trained on a few million image-text pairs to billion-parameter models trained with billions of examples (Wang et al., 2021; Yu et al., 2022; Wang et al., 2022; Chen et al., 2022; 2023). The prohibitive cost of such end-to-end (pre)training, however, has resulted in increased interest in efficient modular methods that leverage existing large language models (LLMs). These align the output of a pretrained image encoder to the LLM's input representation space (Tsimpoukelli et al., 2021; Alayrac et al., 2022; Li et al., 2023a), thereby resulting in a Vision-LLM.

Pretraining vision-language models from scratch requires a massive amount of high-quality image-text data, which is only available in English. Because of this, multilingual pretraining of vision-language models (Ni et al., 2021; Zhou et al., 2021; Zeng et al., 2023; Shan et al., 2022; Li et al., 2023c) commonly supplements limited-size multilingual image-text data with multilingual text-only data (the amount of which often surpasses that of image-text data) to achieve strong results, despite initialization with weights of multilingual text encoders such as XLM-R (Conneau et al., 2020).

In this work, we recognize modular Vision-LLM methods as a potential solution to this problem, observing that: (1) once an image encoder is aligned to one LLM, it requires significantly less data to re-align it to another LLM (Zhang et al., 2023; Zhu et al., 2023) and (2) since image encoding

is, in principle, language-agnostic, it may be possible to successfully re-align the image encoder to a strong multilingual LLM, even if it was initially aligned only with English image-text data. Based on these observations, we present mBLIP, the first massively multilingual modular Vision-LLM, which we obtain by (re-)aligning an image encoder to a multilingual LLM. Putting together a range of recent advances in multimodal representation learning, we efficiently bootstrap a massively multilingual Vision-LLM using only ∼2.5 million images (and without any additional multilingual text-only data), training only 124 million parameters on consumer-grade hardware. We achieve this efficiency by: 1) bootstrapping our model from a) an "English" image encoder (Li et al., 2023a), previously aligned to a monolingual English LLM and b) a strong instruction-tuned multilingual LLM (Xue et al., 2021; Scao et al., 2022; Muennighoff et al., 2022); 2) leveraging recent advances in massively multilingual machine translation (Costa-jussà et al., 2022), which we use to translate high-quality English data—both classic captions as well as task instructions (Dai et al., 2023)—to 95 languages; and finally 3) coupling parameter-efficient training methods (Hu et al., 2022) together with quantization (Dettmers et al., 2022; 2023) to enable training on consumer-grade hardware.

We extensively evaluate mBLIP on different multilingual vision-language tasks to confirm the efficacy of our approach: for multilingual image captioning, mBLIP (with an mT0-XL as the multilingual LLM) surpasses the performance of (zero-shot) PaLI-X (a model with 55B parameters, trained with billions of examples) (Chen et al., 2023) on the XM3600 dataset (Thapliyal et al., 2022). On the visual reasoning and QA tasks of the IGLUE benchmark (Bugliarello et al., 2022), mBLIP matches or surpasses the performance of state-of-the-art models, despite training far fewer parameters on far less pretraining data. Our qualitative analysis demonstrates that mBLIP can handle user input in a wide range of languages and respond appropriately.

## 2 RELATED WORK

### 2.1 LLMS AND IMAGES

The success of scaling up training data and model parameters has resulted in large vision-language models with billions of parameters (Wang et al., 2021; Yu et al., 2022; Wang et al., 2022). However, with the number of parameters in single-digit billions, these are still an order of magnitude smaller than text-only models (Brown et al., 2020); the compute necessary to pretrain comparably large vision-language models, however, is available only to select few (Chen et al., 2022; 2023).

Instead, much of the vision-language research turned to approaches that can leverage the power of existing LLMs with as little training as possible: the most basic approach is to turn the images into textual descriptions (Yang et al., 2021; Liu et al., 2022; Tiong et al., 2022), which already outperforms smaller models on specialized tasks. More sophisticated methods train an image encoder that maps an image into a sequence of tokens in the LLM embedding space (Tsimpoukelli et al., 2021; Alayrac et al., 2022; Li et al., 2023a), while the LLM is kept as-is or is only partially tuned (Alayrac et al., 2022). Most recently, the release of strong publicly available LLMs such as Llama (Touvron et al., 2023) and the success of conversational instruction tuning (Ouyang et al., 2022; Taori et al., 2023; Chiang et al., 2023; Xu et al., 2023), has led to a body of work (Zhu et al., 2023; Liu et al., 2023; Ye et al., 2023; Dai et al., 2023; Gao et al., 2023) that tries to replicate the vision-language skills of GPT-4 (OpenAI, 2023). The vast majority of this research focused on English, for which both an abundance of high-quality image-text data and strong LLMs exist. To the best of our knowledge, we are the first to extend a massively multilingual LLM with "vision capabilities".

### 2.2 MULTILINGUAL VISION-LANGUAGE MODELS

While the majority of research on vision-language models targets English only, a number of multilingual models have been proposed too. M3P (Ni et al., 2021), the first transformer-based (Vaswani et al., 2017) multilingual vision-language model, adopts the architecture and pretraining objectives of English counterparts (Chen et al., 2020; Li et al., 2020). but trains on (i) the code-switched image-text data in which words in English image captions are replaced with translations from various languages as well as (ii) additional (i.e., supplemental) text-only multilingual corpora. UC2 (Zhou et al., 2021) uses a similar architecture and a mix of training objectives but instead of code-switching, it machine translates the 3M captions of CC3M (Sharma et al., 2018) to 5 languages (German, French, Czech, Japanese, and Chinese). Li et al. (2023c) and CCLM (Zeng et al., 2023),

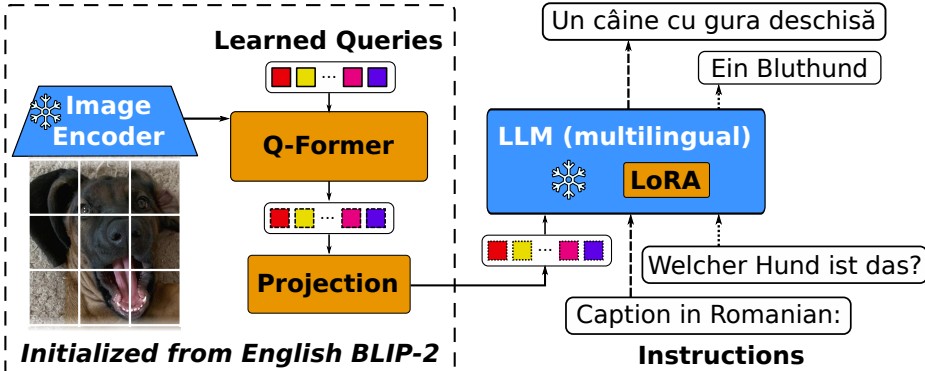

Figure 1: The mBLIP architecture: Following BLIP-2, a Q-Former encodes the image in 32 learned query tokens which are projected to the LLM space. We initialize the Q-Former from an English BLIP-2 models and *re-align* it to the multilingual LLM. Both the image encoder and LLM are frozen during training. Unlike InstructBLIP (Dai et al., 2023) which feeds English instructions into the Q-Former for contextualized tokens, we use LoRA to adapt the LLM in a parameter-efficient fashion.

which adopt the ALBEF architecture (Li et al., 2021) that incorporates additional contrastive learning objectives, use the same translated CC3M data but they additionally supplement 19M parallel sentences (pairing English with all of the languages spanned by their respective downstream evaluation tasks). ERNIE-UniX2 (Shan et al., 2022), with an encoder-decoder architecture, adopts the same pretraining objectives but scales up the data to more translated captions and more text-only data (both aligned and monolingual). Finally, PaLI (Chen et al., 2022) (17B parameters) and PaLI-X (Chen et al., 2023) (55B parameters) represent two huge encoder-decoder models trained using a mixture of vision-and-language tasks, with billions of web-crawled multilingual captions, machine translated data, automatically extracted data (e.g., OCR and object detection), and generated visual QA (VQA) examples. With the exception of the PaLI models and ERNIE-UniX2 – both of which are not publicly available – all other multilingual vision-language models represent encoder-only architectures, which means that they cannot perform image captioning out of the box.

## 3 MBLIP

We first briefly describe the modular BLIP-2 architecture Li et al. (2023a) which we adopt in this work, followed by the description of training tasks and data, which we translate to 95 languages.

### 3.1 ARCHITECTURE

We follow the modular BLIP-2 architecture (Li et al., 2023a) depicted in Figure 1: A Query-Former (Q-Former) is an encoder-only transformer (Vaswani et al., 2017) with 32 learned query tokens as input: it contextualizes the query tokens – via the cross-attention mechanism – with the representations of the image patches encoded by a large (frozen) Vision Transformer (ViT) (Dosovitskiy et al., 2020). The visual tokens that are the output of the Q-Former are then projected into the LLM embedding space with a single linear projection matrix $\mathbf{W}_P \in \mathbb{R}^{h_v \times h_l}$, with $h_v$ and $h_l$ as hidden dimensions (i.e., embedding dimensionality) of the Q-Former and LLM, respectively.

The alignment training on image captioning updates the parameters of the Q-Former (including the 32 query tokens) and the linear projection $\mathbf{W}_P$; all ViT and LLM parameters are kept frozen. Although the Q-Former and projection have initially been aligned to a monolingual English LLM, they only produce *visual* tokens: we believe that as such they are not overly tailored to English and can therefore be effectively re-aligned to a different, multilingual LLM.

Because the LLM is frozen in the BLIP-2 training, its parameters cannot adapt to task-specific idiosyncrasies, e.g., in fine-tuning for VQA or for instruction-following (Dai et al., 2023). Because of this, task-specific fine-tuning of BLIP-2 requires that the text input is not just fed into the LLM but also into the Q-Former in order to enable encoding of *task-specific* visual information from the

input. The pretrained Q-Former, however, is an English-only model, preventing the application of this same approach in the multilingual setting (i.e., we cannot feed the text in other languages into the Q-Former nor efficiently make it massively multilingual, i.e., without a large multilingual pre-training effort). Because of this, we opt for a different approach: instead of feeding the text of the image-text instance (e.g., in VQA) to the Q-Former, we instead allow updates to the LLM. To this effect – given the immense size of LLMs – we resort to parameter-efficient fine-tuning with LoRA (Hu et al., 2022), an approach that trains low-rank reparametrization of the LLM matrices.

## 3.2 Training Tasks and Data

We create a small but high-quality mix of tasks for our re-alignment training. We start from existing high-quality English data and machine-translate it to very many languages in order to obtain multilingual training data for re-alignment of the Q-Former to the multilingual LLM.[1] We hypothesized that the re-alignment to a new LLM can be done with significantly less data than what is needed to train the original Q-Former (Zhu et al., 2023; Zhang et al., 2023). Accordingly, we choose a small but quality mix of English datasets and make it multilingual via MT rather than training with large-scale but very noisy multilingual image-caption datasets like LAION5B (Schuhmann et al., 2022). In addition, in line with findings from language-only instruction-tuning (Sanh et al., 2022; Muennighoff et al., 2022; Chung et al., 2022; Dai et al., 2023; Liu et al., 2023), we expect the training on a mixture of vision-and-language tasks (as opposed to training only for image captioning), with different task instructions, to result in better generalization abilities of the (instruction-tuned) Vision-LLM and improve its downstream performance especially in zero-shot inference.

**Task Mix**: We select below the tasks and datasets we use to create our training data mix for re-alignment (naturally, we ensure that the data does not overlap with our downstream evaluation data; see §4.1). For every task, we create a set of instruction templates with which we generate the training examples (we provide the templates in §C.1 in the Appendix, along with additional details about the training data). In total, across all tasks, we use 5.1M examples encompassing 2.7M unique images.

- **Image Captioning**: We use MSCOCO (Lin et al., 2014) along with 2.3 million examples sampled from the synthetic Web CapFilt dataset (Li et al., 2022). We adopt the noun phrase sampling method from Liu et al. (2023) to ensure concept diversity. Additionally, we use LLaVA-Instruct-Detail (Liu et al., 2023), which contains longer and more detailed captions.

- **Visual Question Answering and Generation**: We use VQAv2 (Goyal et al., 2017) for VQA as well as for the inverse task of question generation (given the answer, the model is supposed to produce the question). Additionally, we split the conversations from LLaVA-Instruct-Conversation into separate VQA pairs. We use A-OKVQA (Schwenk et al., 2022), a knowledge-intensive VQA dataset with rationales behind the answers, to create data for two additional task variants: 1) given the question, generate the answer and the rationale behind it, 2) given the question and the answer, generate the rationale. Finally, we use ImageNet (Deng et al., 2009) with the multilingual labels from Babel-ImageNet (Geigle et al., 2023) framed as an open-ended QA task (with *"What is in the image?"* as the question and without any predefined answer choices).

- **Matching:** Inspired by image-text matching (Lu et al., 2019), where an encoder has to classify if caption and image match, we propose a *yes/no* matching task so that the model learns what is and what is not in the image to reduce hallucinations when interrogating for image content (Li et al., 2023b). For this, we use the Web CapFilt captions for "standard" caption matching with hard negatives. We also use the ImageNet examples with multilingual class labels, where the model has to predict if a given class is in the image or not.

**Machine Translation**: We translate the above English data with NLLB (Costa-jussà et al., 2022) (*nllb-200-distilled-1.3B*), a recent massively multilingual MT model that exhibits strong performance also for low(er)-resource languages. To extend the utility of mBLIP to languages beyond what is covered by existing multilingual evaluation benchmarks, we translate the English data to all languages from the mC4 corpora (Xue et al., 2021),[2] excluding only a handful of languages not

---

[1] Training with only English data, even without LoRA, results in the LLM producing only English output.

[2] https://www.tensorflow.org/datasets/catalog/c4#c4multilingual

supported by NLLB.[3] Our final training dataset thus covers 96 languages (English and 95 translation languages). Translating all English training instances to every target language would result in a 96 times larger dataset (w.r.t. the original English data) and, consequently, prohibitively expensive re-alignment training. We thus translate English instances to target languages in proportion to the languages' representation in mC4 (e.g., we translate 6% of English instances to German, because German represents 6% of the mC4 corpus). We do not translate the answers in A-OKVQA nor most VQAv2 examples[4] because translating short phrases without context is overly error-prone.

To control the output language, we use English prompts that explicitly specify the target language (e.g., *"Answer in French."*). In addition, we translate the instructions for image captioning and LLaVA to the target languages (other templates contain placeholders that make translation difficult).

## 4 EXPERIMENTS

### 4.1 EVALUATION TASKS AND SETUP

We evaluate our model on a range of languages on (1) classification-style VQA and image understanding tasks, where the model generates a short answer in response to a question or premise and (2) image captioning tasks, where the model describes an image. For VQA and image captioning, we ensured that no evaluation instances were used in re-alignment training. In contrast to VQA and image captioning, the model was not exposed to image understanding during re-alignment: these tasks thus test the model's cross-task generalization abilities. To generate outputs, we use beam search with the beam width of 5 and a length penalty of $-1$ for classification-style tasks to encourage short answers. We provide the exact instruction-tuning templates for each task/dataset in §C.2.

**Image Captioning**: XM3600 (Thapliyal et al., 2022) is a captioning dataset covering 36 languages, 3600 images, and ∼2 captions per image and language. xFlickrCo (Bugliarello et al., 2022) combines the 1000 Flickr30k (Plummer et al., 2015) test images with 1000 images from the MSCOCO (Lin et al., 2014) test split[5] and provides one new caption for each image in 8 languages. For the English xFlickrCo results, we use the standard Flickr30k test split (i.e., without MSCOCO images and with 5 reference captions per image). We use CIDEr (Vedantam et al., 2015) as the evaluation metric[6] For Chinese, Japanese, and Thai, which do not use white space for tokenization, we use the default spaCy 3.5.3 segmenter for the respective languages; our results on those languages are thus *not directly comparable* to previous work – which, unfortunately, does not disclose the used tokenizer (Thapliyal et al., 2022; Chen et al., 2022; 2023).

**VQA**: we leverage xGQA (Pfeiffer et al., 2022) and MaXM (Changpinyo et al., 2022), two VQA datasets with 8 and 7 languages, respectively. While answers in xGQA are in English (as only the original GQA (Hudson & Manning, 2019) questions were translated), answers in MaXM are in the language of the question. We evaluate our model in zero-shot inference (i.e., without any additional fine-tuning other than the VQA training included in the re-alignment mix) on both datasets. For xGQA, we additionally fine-tune the model on the training portion of the English GQA and perform cross-lingual zero-shot transfer.[7] We use exact match accuracy with open generation, that is, we do not constrain the generation to a fixed set of labels like, e.g., Zeng et al. (2023). For MaXM, an exact match to any one of the answer candidates is correct, as proposed by Changpinyo et al. (2022).

**Image Understanding**: XVNLI (Bugliarello et al., 2022; Xie et al., 2019) is a visual entailment task that covers 5 languages: given an image and a statement, the model has to decide if the image entails, contradicts or is neutral to the statement. MaRVL (Liu et al., 2021) is based on NLVR2 (Suhr et al., 2019) with new images and concepts spanning different cultures in 6 languages: given two images, the model has to decide if a statement is true or false. We *separately* encode the two images with the

---

[3]Excluded languages (ISO-1/3 codes): *fy*, *haw*, *hmn*, *la*, and *co*.

[4]See §C.1 for details. In short, we limit to the top-1500 answers and use back-translation to increase the likelihood of correct translation. We also still use English half the time.

[5]These MSCOCO captions were created from scratch and not by translating existing MSCOCO captions so this does not constitute leakage from the MSCOCO data of the training mix.

[6]Implementation: `pycocoeval https://github.com/salaniz/pycocoevalcap`

[7]Note that by zero-shot cross-lingual transfer here we refer to the fact that the model has been fine-tuned only on the English GQA data; in re-alignment pretraining, however, it has been exposed to VQA (from other datasets, automatically translated to target languages).

Table 1: Image captioning results (CIDEr) on XM3600 and xFlickrCo for English and averaged over the other languages. **XM3600**: While falling short of the models fine-tuned on MSCOCO translated to all 36 languages (marked by †), our model outperforms training-free methods (LM-Cap) and even PaLI-X 0-shot. Due to different tokenizers for *zh, ja, th*, results are not perfectly comparable. **xFlickrCo**: Our model achieves competitive results compared to English BLIP-2 models. No multilingual baseline on xFlickrCo exists at the time of writing.

| Model | Train P. | Total P. | XM3600 en | XM3600 35-avg |
|---|---|---|---|---|
| Thapliyal et al. (2022) † | 0.8B | 0.8B | 57.60 | 28.90 |
| PaLI-3B † | 3B | 3B | 92.80 | 47.00 |
| PaLI-17B † | 17B | 17B | **98.10** | **53.60** |
| PaLI-X † | 55B | 55B | 94.20 | 53.10 |
| PaLI-X 0-shot | 55B | 55B | 48.80 | 22.70 |
| LMCap (Ramos et al., 2023) | 0 | 3B | 45.20 | 17.60 |
| mBLIP mT0-XL | 124M | 4.9B | 80.17 | 26.77 |
| mBLIP BLOOMZ-7B | 124M | 8.3B | 76.40 | 21.87 |

| Model | Train P. | Total P. | xFlickrCo en | xFlickrCo 7-avg |
|---|---|---|---|---|
| BLIP-2 Flan-T5-XL | 107M | 4.1B | 76.10 | — |
| InstructBLIP Flan-T5-XL | 107M | 4.1B | **84.50** | — |
| mBLIP mT0-XL | 124M | 4.9B | 77.00 | **44.39** |
| mBLIP BLOOMZ-7B | 124M | 8.3B | 76.75 | 42.11 |

Q-Former and then concatenate their visual tokens together as input for the LLM. Like for xGQA, we evaluate the models on XVNLI and MaRVL with (1) zero-shot inference (i.e., no fine-tuning for XVNLI and MaRVL) and (2) supervised cross-lingual transfer: we fine-tune the re-aligned model on the English training portions (of XVNLI and NLVR2, respectively) and evaluate its performance on the test portions of target languages. We report the results in terms of exact match accuracy.

## 4.2 Implementation Details

**Architecture**: We use the BLIP-2 Flan-T5-XL checkpoint to initialize the mBLIP's ViT (EVA CLIP ViT-g/14 (Fang et al., 2022)) and Q-Former. For the multilingual LLM, we experiment with mT0-XL and BLOOMZ-7B (Muennighoff et al., 2022), the instruction-tuned counterparts of mT5-XL (Xue et al., 2021) and BLOOM-7B (Scao et al., 2022), respectively. We use 8-bit (Dettmers et al., 2022) and 4-bit quantization (Dettmers et al., 2023) for the LLM. We merge the LoRA weights obtained in instruction-based re-alignment training into the LLM before we execute LoRA fine-tuning for downstream tasks. We use the HuggingFace Transformers (Wolf et al., 2020) and PEFT[8] libraries for model implementation and LoRA, respectively.

**Training**: We use AdamW (Loshchilov & Hutter, 2019) with weight decay 0.1, learning rate 2e-4 for LoRA and 1e-5 for other parameters; 1000 warm-up steps before a cosine decay; batch size 128 (accomplished via gradient accumulation and checkpointing); we limit the max. target sequence length to 128. For LoRA, which we apply to *all* LLM matrices and not just the query and value matrices of self-attention heads, we set $r = 8, \alpha = 16$ and use dropout with the 0.05 rate. We train on the re-alignment task mixture for 80k steps (2 epochs). Training takes 4 days (mT0) and 6 days (BLOOMZ) with 4 consumer-grade NVIDIA RTX 3090 cards.

**Warmup**: Similar to Zhang et al. (2023), we use a short warmup step before the the re-alignment training where we optimize only the linear projection between the Q-Former and LLM. We use 1M captions to train for 8k steps with a learning rate of 5e-3 (and otherwise the same hyperparameters).

**Fine-tuning**: We train 3 runs (seeds)—reporting their average—for 5/10/20 epochs and batch size 256/128/128 for xGQA/XVNLI/MaRVL, respectively. Other hyperparameters are identical as in re-alignment training. We select the optimal checkpoint based on the English validation data (i.e., we do not use any target-language data for model selection) conforming to requirements of a *true* zero-shot cross-lingual transfer evaluation Schmidt et al. (2022).

## 4.3 Results

**Image Captioning**: Baselines for multilingual image captioning are limited. On the one hand, encoder-based models are usually not evaluated on image captioning, since it is a generative task (Ni et al., 2021; Zhou et al., 2021; Zeng et al., 2023). Decoder-based models, on the other hand,

---

[8]https://github.com/huggingface/peft

Table 2: VQA and image understanding results for English and the average over the other languages: The metric is accuracy with exact matches for the generative models (open generation for mBLIP & PaLI; constrained generation to a set of labels for CCLM on xGQA). Results are averages over 3 fine-tuning runs with different seeds. **Bold** indicates the best score in each column. †: From Zeng et al. (2022b) v1 (arXiv). ‡: Fine-tuned on VQAv2 translated to all MaXM & xGQA languages.

| Model | Train P. | Total P. | xGQA en | 7-avg | XVNLI en | 4-avg | MaRVL en | 5-avg | MaXM en | 6-avg |
|---|---|---|---|---|---|---|---|---|---|---|
| UC2 (Bugliarello et al., 2022) | 270M | 270M | 55.19 | 29.35 | 76.38 | 62.05 | 70.56 | 57.28 | — | — |
| Li et al. (2023c) | 330M | 330M | — | 42.10 | — | 69.50 | — | 62.10 | — | — |
| CCLM (4M) † | 520M | 520M | — | 46.24 | — | 73.32 | 83.22 | 67.17 | — | — |
| CCLM base | 420M | 420M | — | 48.12 | — | 74.78 | — | 68.49 | — | — |
| CCLM large | 970M | 970M | — | **56.25** | — | **78.95** | — | 74.83 | — | — |
| Ernie-UniX2 | 910M | 910M | 56.68 | 45.25 | **87.73** | 77.42 | — | — | — | — |
| Changpinyo et al. (2022) ‡ | 1.5B | 1.5B | 41.50 | 39.44 | — | — | — | — | 36.60 | 42.42 |
| PaLI-17B ‡ | 17B | 17B | 54.20 | 50.77 | — | — | — | — | **56.40** | **57.27** |
| mBLIP mT0-XL (zero-shot) | 124M | 4.9B | 42.55 | 39.20 | 60.61 | 57.65 | 67.26 | 66.66 | 47.99 | 41.04 |
| mBLIP BLOOMZ-7B (zero-shot) | 124M | 8.3B | 43.35 | 37.73 | 58.26 | 55.46 | 62.26 | 58.61 | 55.70 | 27.91 |
| mBLIP mT0-XL (fine-tuned) | 124M | 4.9B | 56.54 | 47.71 | 82.41 | 76.41 | 85.20 | **75.13** | — | — |
| mBLIP BLOOMZ-7B (fine-tuned) | 124M | 8.3B | **57.89** | 44.91 | 75.45 | 66.96 | **86.69** | 73.94 | — | — |

report results only on Chinese (Shan et al., 2022)[9] or are irreproducible because they were trained at scale (compute and data) prohibitive to us (if we were to retrain the models from scratch) and the authors do not publicly release the weights (Thapliyal et al., 2022; Chen et al., 2022).

Table 1 summarizes our image captioning results. On XM3600, mBLIP mT0 outperforms the (training-free) captioning pipeline LMCap (Ramos et al., 2023) as well as PaLI-X (in zero-shot inference): these results are very encouraging, considering that PaLI-X trains orders of magnitude more parameters (55B vs. 124M for mBLIP), on billions of multilingual vision-and-language examples. mBLIP, however, substantially trails the performance of the PaLI models fine-tuned on MSCOCO *with full translations to all 36 languages*: image captioning training on such massive data is, however, prohibitively expensive for the vast majority of researchers and practitioners. While mBLIP is also trained on MSCOCO with translated captions, PaLI models consume orders of magnitude more data in most languages, especially the low-resource ones as they translate *each* English example to *each* of the 36 target languages (yielding from MSCOCO alone $3\times$ more training examples than we do from our entire re-alignment task mix). With proportionally less mBLIP training for lower-resource languages (according to the language-specific corpus portions in mC4), this yields especially large gains for PaLI models for low-resource languages; mBLIP is more competitive for high-resource languages like Spanish or German (see our full results in Table 13).

We additionally evaluate on xFlickrCo. While we are the first to use it for multilingual captioning (in Bugliarello et al. (2022), it is used for image-text retrieval), on the English Flickr30k captions, mBLIP achieves performance that is comparable to that of the (monolingual English) BLIP-2 model.

Finally, between the two mBLIP models, the mT0 variant beats the BLOOMZ variant. We believe this is due to the fact that mT5 (the base LLM from which mT0 was derived) was trained on almost 3 times more text (1 trillion tokens vs. 366 billion) and in nearly twice as many languages as BLOOM (the LLM of BLOOMZ). On a handful of languages like Indonesian or Hindi, however, BLOOMZ outperforms mT0, suggesting that the choice of the mBLIP variant is language-specific.

**VQA and Image Understanding**: Table 2 summarizes the results on VQA and image understanding tasks. On xGQA, mBLIP (zero-shot) outperforms the UC2 model that has been fine-tuned on the GQA data (Zhou et al., 2021; Bugliarello et al., 2022) *for all target languages* (whereas mBLIP was fine-tuned only on English data). When fine-tuned, our mBLIP variants are only outperformed by CCLM (large) (Zeng et al., 2023); CCLM (large) trains nearly nine-times more parameters and leverages more multilingual pretraining data[10]. Crucially, however, CCLM resorts to constrained

---

[9]With the problem that COCO-CN test set (Li et al., 2019) overlaps with the Karpathy's MSCOCO training set (Karpathy & Fei-Fei, 2017), with some captions being translations of English captions. Evaluating mBLIP on COCO-CN would thus represent data leakage, as our re-alignment training uses MSCOCO.

[10]CCLM is also initialized with the English X2-VLM (Zeng et al., 2022a) which is trained on >1B images; the BLIP-2 weights, from which we start the mBLIP training, in contrast, were trained using only 129M images.

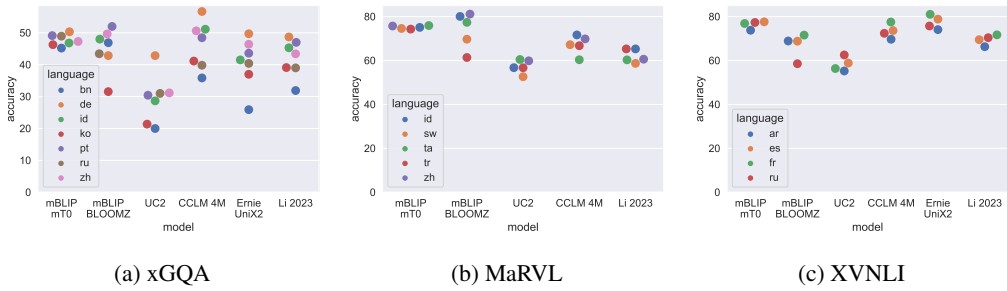

| (a) xGQA | (b) MaRVL | (c) XVNLI |

Figure 2: Cross-lingual transfer results of models fine-tuned on English data on xGQA, MaRVL, and XVNLI. mBLIP mT0 shows a smaller gap between high- and low-resource languages, suggesting better transfer capabilities. (CCLM 4M taken from Zeng et al. (2022b) version 1 on arXiv.)

generation w.r.t. the available answers, which is an easier yet computationally much more demanding evaluation protocol than our open generation. mBLIP exhibits relatively poor zero-shot XVNLI performance, as its LLM fails to predicts the neutral class. After fine-tuning for XVNLI, however, mBLIP mT0 yields multilingual performance (over 4 languages) comparable to that of CCLM (large). The MaRVL zero-shot performance of mBLIP variants is surprisingly good, considering that they were never trained for any task involving multiple images as input; Zero-shot performance of mBLIP mT0 on MaRVL is comparable to that of multiple fine-tuned baselines. When also fine-tuned, mBLIP achieves state-of-the-art MaRVL results, on par with CCLM (large).

On MAXM, mBLIP mT0 (zero-shot) performs comparably to the 1.5B parameter baseline model of Changpinyo et al. (2022) but falls short of the performance of the huge PaLI-17B model. mBLIP BLOOMZ exhibits strong English performance, but surprisingly poor results for other languages. We should emphasize here that training on the translated VQAv2 answers is crucial: without it, the LLM consistently generate answers in English. Even though only ∼25% of examples in have VQAv2 non-English answers, this is already sufficient to eliminate language hallucination (Xue et al., 2021; Vu et al., 2022; Pfeiffer et al., 2023; Li & Murray, 2023), that is, prevent the LLM from generating answers in English regardless of the instruction language[11].

Looking at results for individual languages on the three IGLUE tasks in Figure 2, we see that mBLIP with mT0 greatly improves cross-lingual transfer over prior work, especially for lower-resource languages: while CCLM and Ernie-UniX2 exhibit a gap of 20-25% on xGQA between the best and worst language (German and Bengali), the same gap is only 5% for our fine-tuned mBLIP. Similarly, on MaRVL, CCLM has a gap of 11% between Indonesian and Tamil, while the largest gap for mBLIP amounts to 2%. The same holds for XVNLI, but to a lesser degree: the largest gap between languages for mBLIP (mT0) is 4%, compared to 8% for CCLM/Ernie-UniX2. The BLOOMZ-based variant, however, exhibits much weaker transfer ability and has in fact larger gaps than prior work; this highlights the importance of deriving mBLIP from a strong multilingual LLM.

## 5 ABLATION

We perform ablation experiments for the various components and design decisions for mBLIP, namely: 1) using our instruction mix compared to the 'classic' setting used for BLIP-2 with only image-caption data (using the 2M Web CapFilt examples as training data) and compared to the instruction mix translated following the mT5 language distribution, 2) using LoRA on (all) LLM matrices to better align the LLM to the visual input, and 3) using the warm-start where the projection between Q-Former and LLM is trained briefly in a preliminary stage before the full re-alignment training. We use the zero-shot results on xGQA, XVNLI, xFlickrCo, and XM3600 for evaluation along with fine-tuned results on xGQA. Results are shown in Table 3. In §B, we provide an additional ablation that investigates the effect of adding the matching tasks to re-alignment mix, demonstrating their effectiveness in reducing hallucinations.

---

[11]Training with only English VQAv2 answers during re-alignment results in an mBLIP mT0 instances that achieves only 15.5% accuracy for 6-avg, due to the LLM predominantly generating English answers.

Table 3: Ablations for mBLIP (mT0) w.r.t.: (i) instruction mix (✓) vs. only captions (✗) (i.e., the 2M Web CapFilt examples) vs. instruction mix using the mT5 distribution (mT5), (ii) LoRA (no LoRA ✗, standard LoRA on query&value matrices, LoRA on all matrices), and (iii) using the warm-start where the projection between Q-Former and LLM is trained alone first. All model variants are trained (i.e., re-aligned) for 30k steps.

| Instruction Mix | LoRA | Warm-start | xGQA | | XVNLI | | xFlickrCo | | XM3600 | | xGQA (finetune) | |
|---|---|---|---|---|---|---|---|---|---|---|---|---|
| | | | en | avg | en | avg | en | avg | en | avg | en | avg |
| ✗ | ✗ | ✓ | 26.92 | 9.43 | 34.17 | 35.26 | 77.84 | 37.85 | **86.78** | 22.01 | **56.68** | 46.50 |
| ✗ | all | ✓ | 1.51 | 0.00 | 33.04 | 25.72 | **79.14** | 33.05 | 85.53 | 24.69 | 56.55 | 44.78 |
| ✓ | ✗ | ✓ | 37.33 | 33.77 | 52.02 | 54.26 | 75.92 | 37.86 | 84.14 | 21.35 | 55.72 | 45.36 |
| ✓ | q,v | ✓ | 39.83 | 36.50 | 57.91 | 55.22 | 75.56 | 38.98 | 81.45 | 23.46 | — | — |
| ✓ | all | ✗ | 40.89 | 37.88 | 57.74 | 54.50 | 74.94 | 39.83 | 80.68 | 24.38 | — | — |
| ✓(mT5) | all | ✓ | 40.91 | 37.67 | 58.00 | 54.96 | 72.62 | 39.69 | 80.13 | 25.85 | — | — |
| ✓ | all | ✓ | **41.98** | **38.46** | **58.87** | **56.28** | 77.02 | **40.43** | 81.51 | **25.02** | 56.47 | **46.84** |

**Design:** For zero-shot xGQA and XVNLI, our complete mBLIP configuration yields the best performance. Not using LoRA (i.e., preventing any updates to the LLM) as well as training only on image captioning (compared to the full instruction task mix) both lead to substantially worse performance. Moreover, training (with LoRA) only for image captioning results in a model that does not follow instructions but merely generates captions, making it (zero-shot) useless for other tasks, barring task-specific fine-tuning. For image captioning, both the warm-start and LoRA fine-tuning boost the performance. Unsurprisingly, the re-alignment on captioning alone yields similar or slightly better captioning performance (xFlickCo, XM3600) compared to re-alignment based on the full task mix (i.e., other tasks in the mix do not contribute to captioning ability of mBLIP)[12]. While the task mix brings additional quality captions from MSCOCO and LLaVA (in addition to the Web CapFilt examples), the model also has to learn the other tasks; Importantly, the ablation shows that including other tasks to re-alignment training does not harm the captioning abilities of the model. Finally, looking at supervised xGQA fine-tuning, we observe that all variants exhibit similar performance, regardless of the instruction-tuning (i.e., re-alignment) design. The variants re-aligned only via captioning (first two rows of Table 3) yield even slightly better results than the variants for which VQA was included in the re-alignment training. Contradicting the findings of Dai et al. (2023), our results suggest that more 'complex' instruction-based re-alignment involving a multitude of tasks brings limited gains (if any) for downstream task with large fine-tuning data.

**Language Distribution:** Our translation, proportional to the mC4 language distribution, results in 44% examples in English and, e.g., only 0.003% Lao examples. To test how the language distribution affects performance, we adopt another distribution: that of the mT5's pretraining corpus (reduces English to 8% and pushes Lao to 0.3%). As expected, re-aligning on such a distribution reduces the performance for higher-resource languages, and improves it for low(er)-resource languages. However, the changes in performance (compared to the original mC4 language distribution) are relatively small. This would suggest that it is the language distribution of the (much larger) multilingual pretraining of the LLM that determines the downstream performance for individual languages rather than the language distribution of our (much smaller) re-alignment training.

## 6 CONCLUSION

In this work, we presented mBLIP, the first modular and massively multilingual vision-language model based on a pretrained multilingual LLM. Using a small task mix from quality English datasets, made massively multilingual by means of MT, we re-align an English BLIP-2 model to an instruction-tuned multilingual LLM. Our approach is highly efficient in compute and data requirements and – using recent engineering advances such as 8-bit quantization – can be trained in a few days on consumer-grade hardware (e.g., NVIDIA RTX 3090 cards). We extensively evaluate mBLIP on multilingual vision-language tasks covering image captioning, visual QA, and image understanding to confirm the efficacy of our approach. Results render mBLIP comparable or better than state-of-the-art multilingual vision-language models, despite the fact that we train only a fraction of their number of parameters and on far less data.

---

[12]The captioning-only re-alignment with LoRA failed for xFlickrCo in Russian for unknown reasons, dragging the average down; for other languages the performance is comparable to the full mBLIP configuration.

## 7 ETHICS STATEMENT

While mBLIP is *theoretically* massively multilingual with support for ∼100 languages, our quantitative and qualitative (§A) evaluation shows clear differences between high- and low-resource languages: Image captioning in low-resource languages is more likely to produce incorrect text; Q&A or reasoning prompts show similar results. In most evaluation settings, we further only use English task descriptions and prompts. While our qualitative analysis suggests that the model can handle instructions in other languages, it is likely that English instructions produce better results overall.

Finally, as a general limitation, our models inherit all biases learned by the LLM and the image encoder. The image encoder in particular is trained with mostly English data (that is, images with English captions) and is thus likely best suited for the Anglosphere and might be unable to sufficiently encode concepts (animals, plants, clothing, buildings, etc.) from the various countries and cultures that the LLM supports.

## 8 REPRODUCIBILITY STATEMENT

Our model is initialized with publicly available models. Our training data also uses public datasets and the machine translation is performed with a public model. We detail our training and evaluation setup in §4, and present our training and evaluation templates used to generate examples in §C.

Additionally, we release our code, trained model (re-aligned models only, not the models finetuned on IGLUE tasks), and training data for others to use.

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

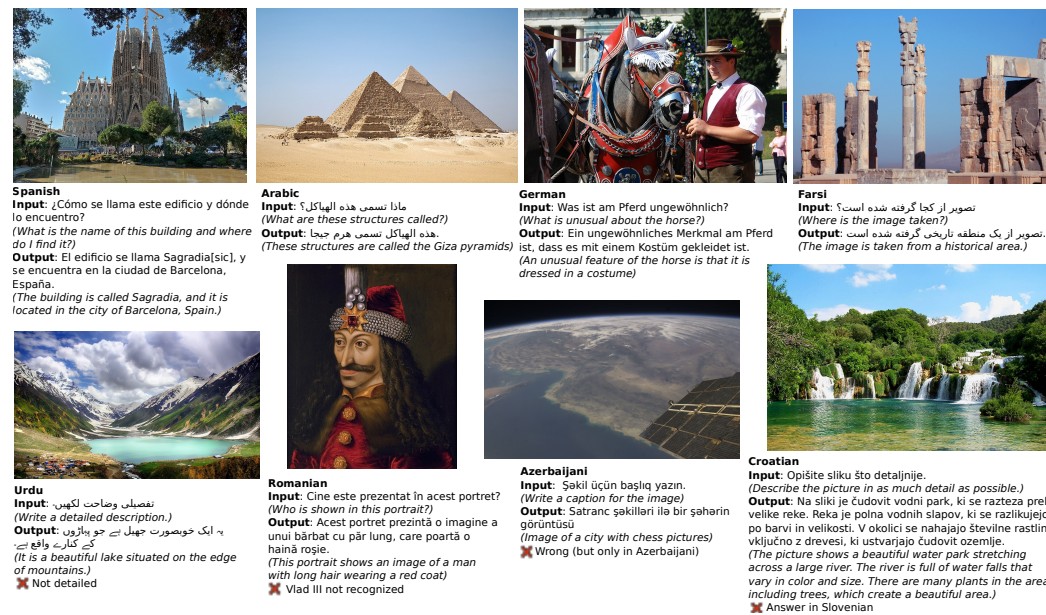

Figure 3: Multilingual examples (with translations from Google Translate in parentheses). While the first row shows that the model can handle captioning and QA in diverse languages, the second row shows some failure cases. We use beam search (5 beams) with a repetition penalty of 1.5.

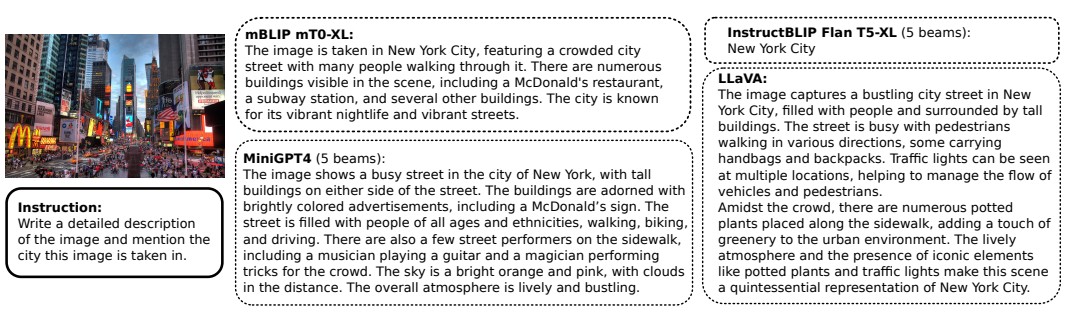

Figure 4: Comparison of English description generated by our model against English models. While our model is less verbose than MiniGPT-4 and LLaVA, it still generates a reasonable answer. InstructBLIP only answers the second part of the instruction (though by varying generation parameters, it is possible to get longer answers but it is not as straightforward). We note that all models struggle with hallucinations in longer descriptions. mBLIP and InstructBLIP examples are generated using the same parameters as in Figure 3, MiniGPT-4 and LLaVA use their official demos.

## A  QUALITATIVE ANALYSIS

In addition to the quantitative evaluation on multilingual datasets of previous sections, we perform a qualitative analysis to better understand the model's visual and multilingual capabilities. As shown in Figure 3, our model can understand instructions in a wide range of languages and describe diverse images, perform simple reasoning, and correctly ground images to world knowledge in those languages. We also see some limitations. The capabilities decrease notably for lower-resource languages. The Urdu example is only a short sentence despite asking for a detailed description. Similarly, the Azerbaijani caption is completely incorrect (and non-sensical), while the model produces a meaningful caption for that same image in many other languages. The Romanian example shows the limitations of the model's world knowledge as the famous portrait of Vlad III is not recognized (neither when asked in Romanian nor in English with various prompts). Finally, the Croatian example shows the difficulty with controlling the output language that we also saw in the quanti-

Table 4: Effect of decision tasks on object hallucination evaluated with POPE (Li et al., 2023b) and CHAIR (Rohrbach et al., 2018) metrics. POPE results improve because the yes-bias is reduced but CHAIR metrics for both short and long captions barely decrease (lower is better).

| | POPE | | | | | | CHAIR | | | |
| | random | | popular | | adversarial | | short | | long | |
| | acc | yes | acc | yes | acc | yes | $C_i$ | $C_s$ | $C_i$ | $C_s$ |
|---|---|---|---|---|---|---|---|---|---|---|
| without matching | 71.00 | 74% | 70.40 | 75% | 63.70 | 81% | 3.10 | 4.50 | 14.90 | 54.70 |
| with matching | 87.30 | 48% | 83.30 | 52% | 76.10 | 59% | 2.40 | 3.50 | 14.10 | 50.50 |

tative evaluation: despite being asked in Croatian, the model answers in (related but still distinct) Slovenian.

In Figure 4, we compare our model's English output against other English Vision-LLMs. While our model is less verbose, it nevertheless can generate reasonable output and correctly follow instructions. We note that mBLIP is prone to hallucination in longer descriptions, as the example shows, but this is a problem that plagues all English Vision-LLM models.

## B    ABLATION: MATCHING TASKS AND OBJECT HALLUCINATIONS

We introduce the matching tasks with the aim of reducing object hallucinations. We evaluate the effectiveness of the measure using two hallucination metrics for English: POPE (Li et al., 2023b) uses interrogative questions ("Is there X in the image?") with random, popular, and adversarial negative objects (using MSCOCO images and object annotations), reporting accuracy and the portion of 'yes' answers due to a yes-bias in most models. CHAIR (Rohrbach et al., 2018) generates captions from MSCOCO images (we use 1k images from the validation split) and then counts hallucinated objects using MSCOCO object annotations. They report the ratio of hallucinated object instances $C_i$, that is of all occurring objects, how many are hallucinated, and the ratio of sentences with hallucinations $C_s$. We generate both short (Prompt: *Caption in English:*) and long captions (Prompt: *Describe the image in English with as much detail as possible.*). We train two models for 30k steps with and without the matching tasks and report results in Table 4. The matching tasks greatly improve results for POPE as they reduce the yes-bias but CHAIR metrics decrease only slightly. This seems to indicate that while matching tasks help for the interrogative POPE questions, they do not noticeably decrease hallucinations when generating captions.

## C    TRAINING AND EVALUATION DATA AND TEMPLATE DETAILS

### C.1    TRAINING

We present our instruction mix in more detail with Table 5 listing the datasets with additional information, and Table 6 listing the templates used to generate the examples.

### C.2    EVALUATION

We present the templates used for the different evaluation datasets in Table 7. Templates for XVNLI and MaRVL are selected using English validation zero-shot performance. XVNLI templates are based on Muennighoff et al. (2022).

We use the same templates for training and inference.

Table 5: Detailed information about the datasets used for training. †: Dataset uses MSCOCO images.

| Dataset | Tasks | #Images | #Examples | Details |
|---|---|---|---|---|
| Web CapFilt (Li et al., 2022) | Image captioning | 2.27m | 2.27m | Subset of the CC3M+CC12M+SBU Web CapFilt dataset[13]. Like Liu et al. (2023), we use spaCy to extract noun phrases and then sample from every phrase with at least 10 occurrences at most 30 captions for a subset covering diverse concepts. |
| | Caption Matching | 600k | 600k | Subset of our image captioning data. We use the CLIP ViT-L/14 by Gadre et al. (2023) to encode images and text to find similar examples for hard negatives. We match every image randomly with the correct caption (50% of the time) or with equal probability a random caption or the 3/10/30/100/300 most similar caption for a mix of very hard to random negatives. |
| MSCOCO (Lin et al., 2014) | Image Captioning | 83k† | 414k | Karpathy training split of MSCOCO (Karpathy & Fei-Fei, 2017) with 5 captions per image. |
| VQAv2 (Goyal et al., 2017) | VQA, VQG | 83k† | 2×443k | Question-answer pairs with ∼5 questions per image. For VQA and VQG, each example is translated to a different language to increase language diversity. We use Google Translate to translate the most common 1500 answers to the 95 languages. We then back-translate them to English and keep only the translations where the back-translation is the original answer; this is to ensure that the answer is (likely) translated correctly. We randomly use either the translated or English answer when generating examples. 83k of the 443k examples have non-English answers. |
| A-OKVQA (Schwenk et al., 2022) | Rational generation, VQA with rational | 11k† | 2×33k | Knowledge-intense VQA questions with additional answer rationals. We generate examples for all three given rationales. We only use the subset of the training split overlapping with the MSCOCO training split. A-OKVQA examples are not translated to any language. |
| LLaVA (Liu et al., 2023) detail | Image captioning | 23k† | 23k | Subset of LLaVA instructions with detailed multi-sentence image captions. |
| LLaVA (Liu et al., 2023) conversations | VQA | 56k† | 219k | Subset of LLaVA instructions with multi-turn dialog; we split the dialogs into independent pairs and keep all pairs with an answer length of max. 3 sentences. |
| ImageNet (Deng et al., 2009) and Babel-ImageNet (Geigle et al., 2023) | VQA | 300k | 300k | Image classification framed as open-ended VQA tasks (i.e., no answer options are given). Babel-ImageNet provides partial translations of the ImageNet classes to the 95 languages. We select one image for every class+language combination (that is, we do not use the full training set). |
| | Matching | 300k | 300k | The model has to decide if a given ImageNet class is correctly in the image. We use the correct label or a random label with equal probability. This uses the same images as the VQA examples but shuffles the image-language pairs. |
| Total | | 2.65m | 5.1m | |

Table 6: Templates used for the training examples. For each example, we randomly select one template. LLaVA examples are used as is since they are already in instruction form. †: Template is translated to the 95 languages.

| Task | Templates |
|------|-----------|
| Image Captioning | Caption the image in $LANGUAGE. |
| | Short $LANGUAGE image caption: |
| | Image caption (in $LANGUAGE): |
| | Briefly describe the image in $LANGUAGE. |
| | Write a short $LANGUAGE image description. |
| | Summarize the image in $LANGUAGE. |
| | Caption the image.† |
| | Short image caption:† |
| | Briefly describe the image.† |
| | Write a short image description.† |
| | Summarize the image.† |
| Caption Matching Question — Yes Answer — No Answer | Does "$CAPTION" accurately describe the image? — Yes, it does. — No, it does not. |
| | Does the caption "$CAPTION" fit the picture? — Yes, it does. — No, it does not. |
| | Does "$CAPTION" correctly summarize the image? — Yes, it does. — No, it does not. |
| | Is "$CAPTION" a good image description? — Yes, it is. — No, it is not. |
| | Is "$CAPTION" a correct caption for the picture? — Yes, it is. — No, it is not. |
| | Is the caption "$CAPTION" a good match for the image? — Yes, it is. — No, it is not. |
| | Decide if the following caption accurately describes the image: $CAPTION. Answer: — Yes, it does. — No, it does not. |
| | Is this caption a good match for the picture? $CAPTION. Answer: — Yes, it is. — No, it is not. |
| | Decide if this caption is a correct summary of the image: $CAPTION. — Yes, it is. — No, it is not. |
| | Would "$CAPTION" be a good image summary? — Yes, it would. — No, it would not. |
| | Would the caption "$CAPTION" fit the picture? — Yes, it would. — No, it would not. |
| | Could you use "$CAPTION" as a caption for the image? — Yes, you could. — No, you could not. |
| VQA | $QUESTION. Short English answer: |
| | Question: $QUESTION. Brief answer (in $LANGUAGE): |
| | Give a short answer in $LANGUAGE to the following question. $QUESTION |
| | Answer the provided question in $LANGUAGE with three words or less. $QUESTION |
| | What is the $LANGUAGE answer to this question? $QUESTION |
| | Briefly answer in $LANGUAGE. $QUESTION |
| VQG | Given the image, generate a question in $LANGUAGE whose answer is: $ANSWER. Question: |
| | Based on the image, create a question (in $LANGUAGE) for which the answer is "$ANSWER". |
| | From the image provided, come up with a $LANGUAGE question that leads to the reply: $ANSWER. Question: |
| | What is a $LANGUAGE question for the image with the answer "$ANSWER"? |
| | Given the image, what would be a $LANGUAGE question that has as answer "$ANSWER"? |
| VQA with rational (instruction templates) | Reason the answer to the following question. $QUESTION |
| | Use reasoning to come to an answer for this question. $QUESTION |
| | Think step-by-step to answer this question. $QUESTION |
| | Answer the following question and explain your answer. $QUESTION |
| | $QUESTION What is the answer and why? |
| VQA with rational (label templates) | $ANSWER. So the answer is $RATIONAL |
| | $ANSWER so $RATIONAL |
| | $RATIONAL. This means the answer is $ANSWER |
| | The answer is $ANSWER because $RATIONAL. |
| | $ANSWER because $RATIONAL. |
| Rational Generation | Question: $QUESTION Answer: $ANSWER. Explanation: |
| | Question: $QUESTION: Answer: $ANSWER. The reason is because |
| | The answer to the question "$QUESTION" is "$ANSWER". Why? |
| | Why is the answer to the question "$QUESTION" "$ANSWER"? |
| | Explain why the answer to the question "$QUESTION" is "$ANSWER" |
| ImageNet Classification | What is the main focus of the image? Short $LANGUAGE answer: |
| | What is in the image? Answer briefly in $LANGUAGE. |
| | This is an image of what? Answer briefly in $LANGUAGE. |
| | What is the central object in the image? Give a short $LANGUAGE answer. |
| | The focus of the image is on what? Short $LANGUAGE answer: |
| | Question: This is an image of what? Answer briefly in $LANGUAGE. |
| | What is at the center of this picture? Short $LANGUAGE answer: |
| | Give a short answer in $LANGUAGE to the following question. What is the main thing shown in the image? |
| | Complete the sentence in $LANGUAGE. This is a photo of a |
| | Name the main thing of this photo in $LANGUAGE: |
| | In less than 3 words in $LANGUAGE, what can be seen in this image? |
| ImageNet Matching Question — Yes Answer — No Answer | Does this image show a $LABEL? — Yes, it does. — No, it does not. |
| | Is there a $LABEL? — Yes, there is. — No, there is not. |
| | Are there any $LABEL in the picture? — Yes, there are. — No, there are not. |
| | Does the image contain a $LABEL? — Yes, it does. — No, it does not. |
| | Yes or no, there is a $LABEL in the photo. — Yes — No |
| | Yes or no, there is a $LABEL visible in the image. — Yes — No |
| | Does this picture have a $LABEL in it? — Yes, it does. — No, it does not. |
| | Can you see a $LABEL in the image? — Yes, you can. — No, you can not. |

Table 7: Templates used for evaluation. XVNLI labels 'entailment', 'contradiction', and 'neutral' are remapped to 'yes', 'no', 'maybe', respectively; MaRVL labels 'true' & 'false' are remapped to 'yes', 'no', respectively.

| Dataset | Template |
|---------|----------|
| xFlickrCo, XM3600 | Caption in $LANGUAGE: |
| xGQA, MaXM | Question: $QUESTION Short answer in $LANGUAGE: |
| XVNLI | Is it guaranteed true that "$HYPOTHESIS"? Yes, no, or maybe? Answer in English: |
| MaRVL | Based on the two images, is it correct to say "$STATEMENT"? Yes or no? Answer in English: |

## D    IMAGE ATTRIBUTION

Image attribution for Figure 3 in order of appearance from top-left to bottom-right:

- Sagrada Familia: `https://de.wikipedia.org/wiki/Datei:Sagrada_Familia_8-12-21_(1).jpg`. Canaan, CC BY-SA 4.0 `https://creativecommons.org/licenses/by-sa/4.0`, via Wikimedia Commons

- Giza: `https://commons.wikimedia.org/wiki/File:All_Gizah_Pyramids.jpg`. Ricardo Liberato, CC BY-SA 2.0 `https://creativecommons.org/licenses/by-sa/2.0`, via Wikimedia Commons

- Oktoberfest Kutsche: `https://de.wikipedia.org/wiki/Datei:Oktoberfest-Kutscher.jpg`. Hullbr3ach, CC BY-SA 2.5 `https://creativecommons.org/licenses/by-sa/2.5`, via Wikimedia Commons

- Gate of All Nations, Persepolis: `https://commons.wikimedia.org/wiki/File:Gate_of_All_Nations,_Persepolis.jpg`. Alborzagros, CC BY-SA 3.0 `https://creativecommons.org/licenses/by-sa/3.0`, via Wikimedia Commons

- Lake saif ul malook: `https://en.wikipedia.org/wiki/File:Lake-saif-ul-malook_Pakistan.jpg`. Ayesha.great, CC BY-SA 4.0 `https://creativecommons.org/licenses/by-sa/4.0`, via Wikimedia Commons

- Vlad III: `https://en.wikipedia.org/wiki/File:Vlad_Tepes_002.jpg`. Portrait of Vlad III the Impaler

- Satellite: `https://en.wikipedia.org/wiki/File:Jaz_Murian_satellite.jpg`. NASA, Public domain, via Wikimedia Commons

- Krk waterfalls: `https://commons.wikimedia.org/wiki/File:Krk_waterfalls.jpg`. Version13 at English Wikipedia, Public domain, via Wikimedia Commons

Image attribution for Figure 4: `https://commons.wikimedia.org/wiki/File:New_york_times_square-terabass.jpg`. Terabass, CC BY-SA 3.0 `https://creativecommons.org/licenses/by-sa/3.0`, via Wikimedia Commons

## E    FULL RESULTS

Table 8: Results in all languages for xGQA. Finetuned results are averaged over 3 seeds.

|  | bn | de | id | ko | pt | ru | zh |
|---|---|---|---|---|---|---|---|
| mBLIP mT0-XL (zero-shot) | 38.51 | 40.53 | 38.34 | 38.31 | 40.15 | 39.59 | 38.99 |
| mBLIP mT0-XL (finetuned) | 45.21 | 50.32 | 46.80 | 46.28 | 49.12 | 48.94 | 47.28 |
| mBLIP BLOOMZ-7B (zero-shot) | 38.96 | 37.04 | 39.99 | 29.06 | 41.78 | 37.55 | 39.72 |
| mBLIP BLOOMZ-7B (finetuned) | 46.90 | 42.86 | 48.01 | 31.56 | 51.99 | 43.44 | 49.64 |

Table 9: Results in all languages for XVNLI. Finetuned results are averaged over 3 seeds.

|  | ar | es | fr | ru |
|---|---|---|---|---|
| mBLIP mT0-XL (zero-shot) | 56.26 | 57.57 | 58.52 | 58.26 |
| mBLIP mT0-XL (finetuned) | 73.80 | 77.62 | 76.87 | 77.33 |
| mBLIP BLOOMZ-7B (zero-shot) | 56.26 | 56.17 | 57.74 | 51.65 |
| mBLIP BLOOMZ-7B (finetuned) | 68.90 | 68.81 | 71.57 | 58.55 |

Table 10: Results in all languages for MaRVL. Finetuned results are averaged over 3 seeds.

|  | id | sw | ta | tr | zh |
|---|---|---|---|---|---|
| mBLIP mT0-XL (zero-shot) | 64.89 | 64.80 | 69.65 | 68.05 | 65.91 |
| mBLIP mT0-XL (finetuned) | 75.09 | 74.61 | 75.93 | 74.32 | 75.72 |
| mBLIP BLOOMZ-7B (zero-shot) | 59.13 | 56.23 | 60.31 | 57.71 | 59.68 |
| mBLIP BLOOMZ-7B (finetuned) | 80.08 | 69.71 | 77.38 | 61.38 | 81.16 |

Table 11: Results in all languages for MaXM.

|  | fr | hi | iw | ro | th | zh |
|---|---|---|---|---|---|---|
| mBLIP mT0-XL (zero-shot) | 40.61 | 48.30 | 35.56 | 41.74 | 53.97 | 26.06 |
| mBLIP BLOOMZ-7B (zero-shot) | 22.87 | 52.38 | 18.41 | 31.83 | 17.22 | 24.76 |

Table 12: Results in all languages for xFlickrCo.

|  | de | es | id | ja | ru | tr | zh |
|---|---|---|---|---|---|---|---|
| mBLIP mT0-XL (zero-shot) | 58.23 | 64.86 | 47.44 | 33.27 | 41.77 | 35.18 | 29.98 |
| mBLIP BLOOMZ-7B (zero-shot) | 50.50 | 64.89 | 54.42 | 29.10 | 38.36 | 25.08 | 32.42 |

Table 13: Results in all languages for XM3600.

|  | ar | bn | cs | da | de | el | es | fa | fi | fil | fr | he |
|---|---|---|---|---|---|---|---|---|---|---|---|---|
| mBLIP mT0-XL (zero-shot) | 21.13 | 11.30 | 31.84 | 44.19 | 32.48 | 23.36 | 62.61 | 0.00 | 16.78 | 17.71 | 57.64 | 18.69 |
| mBLIP BLOOMZ-7B (zero-shot) | 27.78 | 16.12 | 21.77 | 25.25 | 30.04 | 14.12 | 60.03 | 13.84 | 4.69 | 1.99 | 60.42 | 7.16 |

|  | hi | hr | hu | id | it | ja | ko | mi | nl | no | pl | pt |
|---|---|---|---|---|---|---|---|---|---|---|---|---|
|  | 16.07 | 5.18 | 21.54 | 38.53 | 45.19 | 33.23 | 10.39 | 4.09 | 55.72 | 46.15 | 31.22 | 53.13 |
|  | 24.91 | 2.13 | 10.99 | 45.29 | 42.40 | 25.43 | 2.54 | 0.02 | 45.54 | 25.01 | 20.65 | 47.79 |

|  | quz | ro | ru | sv | sw | te | th | tr | uk | vi | zh |  |
|---|---|---|---|---|---|---|---|---|---|---|---|---|
|  | 1.08 | 21.71 | 27.25 | 48.38 | 11.76 | 11.20 | 41.93 | 22.64 | 0.00 | 39.24 | 13.48 |  |
|  | 0.02 | 17.62 | 22.83 | 31.77 | 8.45 | 8.65 | 8.16 | 14.21 | 8.97 | 54.29 | 14.65 |  |

