# OpenReview forum: "mBLIP: Efficient Bootstrapping of Multilingual Vision-LLMs"
_ICLR.cc/2024/Conference — Submitted to ICLR 2024_

### Official Review · Reviewer_Gwix · 2023-10-31

**Soundness:** 3 good
**Presentation:** 3 good
**Contribution:** 3 good
**Rating:** 6
**Confidence:** 4

**Summary:**

This paper proposed an efficient method to adapt a pre-trained English vision language model to multilingual models. The dataset is created from an English only source, with machine translation to obtain multilingual data.The model is then partially finetuned on the machine translated multilingual dataset, where the Q-former in between is trainable and the LLM is LoRA tuned. Experimental results are behind the state-of-the-art methods, but it provides a cheap way to support multilinguality for an English model.

**Strengths:**

1. The proposed method is quite affordable, in that the training only takes a few days on a few GPUs. The method overall is adaptation cheap thanks to the limited amount of trainable parameters in Q-former, and the LoRA tuning of LLMs.
2. This paper tackles an important problem of multilingual vision language models, that enables wider accessibility of the vision language models especially for non-English speakers.

**Weaknesses:**

1. One weakness is to name a translated dataset as a multilingual dataset, which doesn't cover the real benefits of multilingual at all (i.e. multiple cultures, examples from different locations from local people speaking those languages). This is misleading as the future work will use the translated dataset which will reinforce the emphasis of English centric research, under the translated "multilingual" directions.
2. The idea of applying Q-Former to LLMs is not novel and has been explored by the previous BLIP papers. I have not found BLIP baselines reported (at least on EN) in the paper (except for the right sub-table of Table 1). More comparisons with BLIP baselines, especially on the impact of multilingual data to the original English tasks would be nice to show the impact of this paper.

**Questions:**

See above sections. More questions:

1. The comparison of mBLIP with PaLI is quite unclear. Is mBLIP a zero-shot model or finetuned model? If it's the latter then it doesn't make any sense to claim that mBLIP outperforms (zero-shot) PaLI.

---

> ### Author Response · Authors · 2023-11-15
>
> Thank you for your review.
>
> > One weakness is to name a translated dataset as a multilingual dataset, which doesn't cover the real benefits of multilingual at all (i.e. multiple cultures, examples from different locations from local people speaking those languages). This is misleading as the future work will use the translated dataset which will reinforce the emphasis of English centric research, under the translated "multilingual" directions.
>
> We fully agree with the reviewer, but want to point out that this is, unfortunately, a general problem in multilingual vision-language models. For example, CCLM, Ernie-UniX2, or UC2 all also train with machine-translated data. PaLI models are trained with both machine-translated data and (proprietary) multilingual web-crawled data. There is multilingual caption data available in Laion5B but these are noisy web-crawled captions. For high-quality captions and other tasks like VQA, there is only English data available.
>
> However, our evaluation setup does take this problem into account and includes test datasets that require both multilingual and multicultural knowledge: XM3600 in particular selected and annotated images from 36 countries with speakers from those countries, and MaRVL is built from the ground up (and also annotated by native people) to include images and concepts from 5 different countries.
>
> Nonetheless, you make a valid point and we will make sure to add a short discussion on true multilingual data vs. “multilingual” data obtained through translation and disadvantages/pitfalls of the latter.
>
> > The idea of applying Q-Former to LLMs is not novel and has been explored by the previous BLIP papers. I have not found BLIP baselines reported (at least on EN) in the paper (except for the right sub-table of Table 1). More comparisons with BLIP baselines, especially on the impact of multilingual data to the original English tasks would be nice to show the impact of this paper.
>
> The comparison with BLIP-2 models is difficult because their underlying English LLMs are better on English tasks than multilingual models. Monolingual language-specific LLMs (especially if the language is English) generally outperform multilingual counterparts for that particular language. This phenomenon (i.e., the downside of sharing representation space across many languages) is known as the curse of multilinguality [1,2].
>
> We did consider including the zero-shot GQA results for BLIP-2 and InstructBLIP Flan-T5-XL results in Table 2 (44.0 and 48.4% accuracy, our mBLIP mT0-XL has 42.6) but decided against it for the sake of readability of Table 2, which is already quite full with information.
>
> > The comparison of mBLIP with PaLI is quite unclear. Is mBLIP a zero-shot model or finetuned model? If it's the latter then it doesn't make any sense to claim that mBLIP outperforms (zero-shot) PaLI.
>
> This is a good question with no straightforward answer:
>
> PaLI-X is pre-trained with (amongst other data) millions of captions translated to the 36 XM3600 languages. For fine-tuning, they are then trained on ~5M examples of MSCOCO again translated to the 36 languages.
>
> mBLIP is trained with our task mix for 2 epochs, which includes 400k MSCOCO captions automatically translated to 95 languages.
>
> So on one hand, mBLIP has seen data for some of the downstream fine-tuning tasks in its re-alignment training (but orders of magnitude less than the fine-tuned PaLI) but on the other hand, we did not optimize our setup for the 36 XM3600 languages (and PaLI does).
>
> To make the comparison more complicated, our mBLIP models have 5 to 8B parameters of which we train only 128M. Pali-X has 55B parameters, all of which are updated during fine-tuning training.
>
> In sum, this means that mBLIP has some advantage regarding the (type of the) training data, whereas Pali-X has an advantage in terms of training languages (matching exactly those from XM3600), model size, and size of the (pre)training data. We thus believe that it is overall fair to compare mBLIP against Pali-X (zero-shot).
>
> References:
>
> [1] Conneau, A., Khandelwal, K., Goyal, N., Chaudhary, V., Wenzek, G., Guzmán, F., ... & Stoyanov, V. (2020, July). Unsupervised Cross-lingual Representation Learning at Scale. In Proceedings of the 58th Annual Meeting of the Association for Computational Linguistics (pp. 8440-8451).
>
> [2] Pfeiffer, J., Goyal, N., Lin, X., Li, X., Cross, J., Riedel, S., & Artetxe, M. (2022, July). Lifting the Curse of Multilinguality by Pre-training Modular Transformers. In Proceedings of the 2022 Conference of the North American Chapter of the Association for Computational Linguistics: Human Language Technologies (pp. 3479-3495).

---

> > ### Comment · Reviewer_Gwix · 2023-12-03
> >
> > Thank you for the clarifications. The answers partially addressed my concerns. For the comparison with PaLI 0-shot, I agree it's indeed very tricky. I suggest to not highlight it in the Table 1 caption, and/or mark a star in the table row and just use it for discussion in the main text. 0-shot task is very different from finetuning task, thus any direct comparison in a table would create confusion. I will keep my original borderline accept rating.

---

### Official Review · Reviewer_deKR · 2023-11-01

**Soundness:** 2 fair
**Presentation:** 3 good
**Contribution:** 2 fair
**Rating:** 5
**Confidence:** 4

**Summary:**

The work introduces mBLIP, which is a multi-lingual version of BLIP2. In summary, mBLIP replaces BLIP2's language decoder with stronger and newer multi-lingual LLMs (which breaks the alignment of vision encoder and the LLM) and re-aligns the vision encoder and the LLM with a dataset with images-text pairs, containing 96 machine translated languages.

**Strengths:**

- The work proposes a dataset (if will be released) with image-text pairs coming from 96 different languages, which could be useful for future research.
- Experiments are decent. The work also covers some important ablation studies.

**Weaknesses:**

- The major concern is the novelty. In short, the work can be summarized as (1) using machine translation to translate image-text pairs (in English) to different languages, and (2) replace the language decoder in BLIP2 with a multi-lingual one and train the model with translated dataset.
- The mixture of task highlighted in the paper as a contribution to the success of the model, although not considered in BLIP2, was actually explored in PALI. This fact further deteriorates the novelty of this work.
- It's not a surprise that task mix helps most in VQA and VNLI tasks because such an objective in included in the pre-training stage. The mixture of task actually hurts the captioning performance.
- The model can be trained on 4 cheap RTX-3090 GPUs, which is great. But this claim in the conclusion has nothing to do with the contribution of the work, i.e., any similar models will enjoy the speed-up with 8-bit quantization and LoRA PEFT.

**Questions:**

- The paper does not state the data portion used to pre-train the model. In the model pre-training, 3 tasks are considered, but the captioning dataset looks much bigger than VQAs. Meanwhile, the small size of VQA could contribute less to comprehensive pre-training, but only provide some demonstration to downstream QA tasks (that's probably why QA and VNLI tasks perform well).
- The paper claims that based on a few prior works, the re-alignment does not need larger dataset. However, some of the listed prior works (on arXiv) have very weak quantitative studies, which may not support authors' claim. Thus, such a claim has to be validated by ablation studies like re-alignment with different size of pre-training data, e.g., 3M, 12M, 120M, 400M, etc.

---

> ### Author Response · Authors · 2023-11-15
>
> Thank you for your review.
>
> >  (if will be released)
>
> Yes, all our training data along with all trained models (full weights) will be publicly released.
>
> > The major concern is the novelty. In short, the work can be summarized as (1) using machine translation to translate image-text pairs (in English) to different languages, and (2) replace the language decoder in BLIP2 with a multi-lingual one and train the model with translated dataset.
>
> Please see our general response to all reviewers.
>
> > The mixture of task highlighted in the paper as a contribution to the success of the model, although not considered in BLIP2, was actually explored in PALI. This fact further deteriorates the novelty of this work.
>
> It would appear that we forgot to cite Pali when motivating our task mix in §3.2; we will fix this. We did however cite InstructBLIP, another Vision-LLM trained with a task mix, so we did not intend to claim this as a particular novelty of our approach.
>
> However, unlike Pali, who use a rigid (and English-only) input format for instructions, we use diverse templates and additionally incorporate translated user instruction data (from Llava) so that our mBLIP models can handle a wider range of user inputs as illustrated in Figure 3.
>
> > It's not a surprise that task mix helps most in VQA and VNLI tasks because such an objective in included in the pre-training stage. The mixture of task actually hurts the captioning performance.
>
> When comparing the results in Table 3 (row 2 for captions only, last row for the task mix), we do not see that the task mix hurts captioning performance: while English results for XM3600 and xFlickrCo are slightly lower, multilingual results are higher.
>
> Also, VNLI is not included in the task mix, which is why our model has poor zero-shot results there (in zero-shot, i.e., without fine-tuning for VNLI, it never predicts the neutral class).
>
> > The model can be trained on 4 cheap RTX-3090 GPUs, which is great. But this claim in the conclusion has nothing to do with the contribution of the work, i.e., any similar models will enjoy the speed-up with 8-bit quantization and LoRA PEFT.
>
> First, while LoRA and 8-bit quantization do contribute to the training efficiency, the core driver of our efficient training is the (comparatively small) size of our training data and our task mix – precisely because we re-align a pre-trained “English” Q-former to a new, multilingual LLM, we are able to achieve multilingual performance competitive to that of much larger models that have been trained from scratch with orders of magnitude less training data.
>
> Second, the fact that our approach works with other models should be seen as a positive: if next week, a new SOTA multilingual LLM is released, we have a proven computationally effective method of inducing a corresponding Vision-LLM.
>
> > The paper does not state the data portion used to pre-train the model. In the model pre-training, 3 tasks are considered, but the captioning dataset looks much bigger than VQAs. Meanwhile, the small size of VQA could contribute less to comprehensive pre-training, but only provide some demonstration to downstream QA tasks (that's probably why QA and VNLI tasks perform well).
>
> We do report this. In §3.2, we refer to Appendix C.1 for details of our training data where table 5 lists the exact number of images and samples used from each dataset.
>
> As can be seen there, we limit the caption datasets to prevent this very problem you describe where the larger caption dataset would (proportionally) overshadow the other tasks.
>
> > The paper claims that based on a few prior works, the re-alignment does not need larger dataset. However, some of the listed prior works (on arXiv) have very weak quantitative studies, which may not support authors' claim. Thus, such a claim has to be validated by ablation studies like re-alignment with different size of pre-training data, e.g., 3M, 12M, 120M, 400M, etc.
>
> We cite two works to support this claim. We can agree that Zhu et al., 2023 lacks proper quantitative evaluation; Zhang et al., 2023, however, features extensive experiments that clearly indicate that one can get comparable results with re-alignment on far less data.
>
> While the suggested experiment would be interesting – we do not dispute that with more data, you can (very likely) get better results, but we also suspect that the gains would diminish logarithmically with more data. The proposed data-scaling experiment/ablation is also too expensive for us to run: training with 100M examples would take at least 40 days on our compute.
>
> In addition, there is another problem: for the greater data scale (>100M images), only image-caption data is available, which would greatly skew the proportions in our task mix.

---

> > ### Comment · Reviewer_deKR · 2023-11-22
> >
> > Thanks for the authors' responses. However, some of my concerns remain and I'm not fully convinced by the contributions/novelties of the work. I thus keep my original borderline rating.

---

### Official Review · Reviewer_XtFb · 2023-11-01

**Soundness:** 3 good
**Presentation:** 3 good
**Contribution:** 3 good
**Rating:** 6
**Confidence:** 3

**Summary:**

This paper is to propose mBLIP: Introduction of the first multilingual Vision-LLM, mBLIP, achieved by realigning an image encoder to a multilingual LLM from English only Vision-LLM. In addition, mBLIP is created using only about 2.5 million images and training 124 million parameters on consumer hardware to convert high-quality English data into 95 languages for training.
The contribution of this paper is to presents a cost-effective approach to developing multilingual vision-language models with broad language coverage and strong performance.

**Strengths:**

mBLIP is created using only about 2.5 million images and training 124 million parameters on consumer hardware to convert high-quality English data into 95 languages for training.
The contribution of this paper is to presents a cost-effective approach to developing multilingual vision-language models with broad language coverage and strong performance.

**Weaknesses:**

There are still big differences of accuracy between English and other languages on XM3600 and xFlickerCo testing.
In addition, Table 2 is not so clean to understand.

**Questions:**

There are still big differences of accuracy between English and other languages on XM3600 and xFlickerCo testing.
In addition, Table 2 is not so clean to understand.

---

> ### Author Response · Authors · 2023-11-15
>
> Thank you for your review.
>
> > There are still big differences of accuracy between English and other languages on XM3600 and xFlickerCo testing.
>
> For xFlickrCo, we are comparing against English-only monolingual models based on Flan-T5. Monolingual language-specific LLMs (especially if the language is English, the most resourced language there is) generally outperform multilingual counterparts for that particular language. This phenomenon is known as the curse of multilinguality [1,2].  English-only models are in general better in English than multilingual models so the gap in CIDEr score is to be expected there.
>
> For XM3600, the English CIDEr scores seem to us qualitatively similar to the multilingual results: our mBLIP variants outperform Pali-X (zeroshot), LMCap, and Thapliyal et al. (2022) and are outperformed by the Pali models that have been fine-tuned on MSCOCO.
>
>
> > In addition, Table 2 is not so clean to understand.
>
>
> We are aware that Table 2 contains a lot of information but we tried our best to explain everything clearly in the caption (and the text of discussions associated with the Table). Is there anything in particular that was hard to understand so we adjust the table accordingly?
>
>
> [1] Conneau, A., Khandelwal, K., Goyal, N., Chaudhary, V., Wenzek, G., Guzmán, F., ... & Stoyanov, V. (2020, July). Unsupervised Cross-lingual Representation Learning at Scale. In Proceedings of the 58th Annual Meeting of the Association for Computational Linguistics (pp. 8440-8451).
>
> [2] Pfeiffer, J., Goyal, N., Lin, X., Li, X., Cross, J., Riedel, S., & Artetxe, M. (2022, July). Lifting the Curse of Multilinguality by Pre-training Modular Transformers. In Proceedings of the 2022 Conference of the North American Chapter of the Association for Computational Linguistics: Human Language Technologies (pp. 3479-3495).

---

### Official Review · Reviewer_8jee · 2023-11-01

**Soundness:** 3 good
**Presentation:** 3 good
**Contribution:** 2 fair
**Rating:** 5
**Confidence:** 4

**Summary:**

The paper proposes a multi-lingual LVLM that relies on aligning a frozen backbone and an LLM following BLIP2. The novelty relies on the alignment to a multi-language model, and how to tweak the alignment module (the Qformer) for this use case from a starting point of having a Qformer trained for the English language.

**Strengths:**

The approach is interesting in the sense that it covers a reasonable use case, that of multi-lingual LVLM by aligning frozen models. It seems like the authors found a good space where a solution did not exist in the literature.

The proposed approach seems technically reasonable.

The paper is mostly clear, the target application is clear and the explanations are usually clear (although some issues are flagged later on).

Results are positive, although there isn't a lot to compare against.

**Weaknesses:**

The paper does not seem to have a strong technical contribution. On the pro side, I like the application, and there's some analysis of the impact of the different pieces in Table 3, which offsets somewhat the lack of technical contribution.

I am a bit unsure about the impact of LoRA. A task prompt has much lower capacity to adapt to any downstream task compared to adding LoRA (especially if LoRA adapters are added to every 1x1 layer). So I'm wondering if part of the performance is due to LoRA vs no LoRA.

Also, if there is enough data to train LoRA adapters, why then there is no data to train a task prompt?

is "re-aligning" actually fine-tuning?

**Questions:**

Any answers to the "weakness" above. In particular: did I miss something with regards to novelty? can the use of LoRA be better justified?

---

> ### Author Response · Authors · 2023-11-15
>
> Thank you for your review.
>
> > The paper does not seem to have a strong technical contribution. On the pro side, I like the application, and there's some analysis of the impact of the different pieces in Table 3, which offsets somewhat the lack of technical contribution.
>
> Please see our general response to reviewers.
>
> > I am a bit unsure about the impact of LoRA. A task prompt has much lower capacity to adapt to any downstream task compared to adding LoRA (especially if LoRA adapters are added to every 1x1 layer). So I'm wondering if part of the performance is due to LoRA vs no LoRA.
>
> We are not sure what you refer to with “task prompts”? Do you mean (1) InstructBLIP’s method of including the prompt in the Q-former for task-contextualized visual tokens or (2) prefix-tuning, i.e., the trainable prompt-token embeddings as proposed in [1,2]?
>
> Considering the latter more likely – in the LoRA paper, the authors demonstrate that LoRA significantly outperforms prefix embeddings, as well as the other parameter-efficient fine-tuning (PEFT) approaches, such as bottleneck adapters or BitFit. They also show that it is the only PEFT method that consistently matches (or surpasses) the performance of full fine-tuning. For all these reasons, LoRA is arguably the most widely used PEFT technique in NLP.  LoRA fine-tuning of the multilingual LLM in our approach indeed contributes to the overall performance, as demonstrated by the ablation from Table 3 (row 3 vs. last row): LoRA fine-tuning of the multilingua LLM yields better results than when the LLM is kept frozen.
>
> > Also, if there is enough data to train LoRA adapters, why then there is no data to train a task prompt?
>
> We are somewhat puzzled by this question and are not sure what exactly you refer to. We are not aware of making a claim that there is not enough data to train a task prompt anywhere in the paper.
>
> If by task prompt you mean InstructBLIP’s method with prompt-contextualized visual tokens, then the problem is the one described in §3.1: the Q-former is BERT-based (i.e., a monolingual English Transformer Encoder), not capable of handling multilingual prompts (i.e., prompts in languages other than English).
>
> If (which we believe is more likely) by task prompt you refer to prefix embeddings (prefix-tuning) in the vein of [1,2]: then there is no conceptual obstacle to doing prefix-tuning instead of LoRA, but it would (1) lead to inferior performance to LoRA (as stated above,  LoRA has been documented to outperform competing PEFT methods and is de facto the default PEFT approach at the moment in NLP), and (2) increasing both training and inference time, due to the increase in sequence length (from additional prefix embeddings) and quadratic nature of the attention mechanism in the Transformer. So while prefix-tuning is conceptually viable PEFT technique for our setting (but so are the bottleneck adapters or BitFit), there is no reason to use it over LoRA. We are happy to emphasize this in the manuscript (advantages of LoRA over competing parameter-efficient fine-tuning methods)
>
> > is "re-aligning" actually fine-tuning?
>
> In a way, yes: The (multilingual) LLM is fine-tuned in a parameter-efficient manner with LoRA; the Q-former is first aligned to the LLM (similar to stage two in BLIP-2) and then further fine-tuned with the task mix (similar to the InstructBLIP fine-tuning of a BLIP-2 model) in one training step. However, to better illustrate that our approach re-aligns a Q-former from one LLM to a new (multilingual) LLM and to separate this step from the classic task-specific fine-tuning in our downstream evaluation (for xGQA, XVNLI, & MaRVL) we decided to use ‘re-align’ instead of ‘fine-tune’ for the training on our task mix.
>
> > can the use of LoRA be better justified?
>
> As we motivate in §3.1 and show in the ablation, training the LLM yields better results than keeping it frozen. Full fine-tuning of all parameters of the LLM is too computationally expensive and one must resort to parameter-efficient fine-tuning (PEFT). LoRA is, as discussed above, widely accepted in NLP as the best PEFT technique – outperforming the competing approaches (bottleneck adapters, bitfit, prefix-tuning) and often matching the performance of full fine-tuning.
>
>
> [1] Li & Liang, 2021. Prefix-Tuning: Optimizing Continuous Prompts for Generation
>
> [2] Lester et al., 2021. The Power of Scale for Parameter-Efficient Prompt Tuning

---

> > ### Comment · Reviewer_8jee · 2023-11-21
> > **Discussion by reviewer**
> >
> > Thanks for the response. I think I was not clear in my original review so I'll try to do better this time:
> >
> > 1 - Regarding the original question: "Also, if there is enough data to train LoRA adapters, why then there is no data to train a task prompt?" and the answer "We are somewhat puzzled by this question and are not sure what exactly you refer to. We are not aware of making a claim that there is not enough data to train a task prompt anywhere in the paper."
> >
> > I used the wrong word, sorry. I meant to ask if there is the option of training the Q-Former. The relevant phrase is this:
> > "The pretrained Q-Former, however, is an English-only model, preventing the application of this same approach in the multilingual setting (i.e., we cannot feed the text in other languages into the Q-Former nor efficiently make it massively multilingual, i.e., without a large multilingual pretraining effort)."
> >
> > I was wondering if it would be possible to adapt the Qformer instead of the multi-lingual LLM (through LoRA, FT, other adapters... which specific method was not my point). My understanding right now is: for task-specific FT, task-related textual information (basically a manually-crafted template) has to go through the QFormer, which is mono-lingual, thus creating an issue of misalignment between the output of the QFormer and the specific language being used (if other than English). The authors constructed task-specific data in all languages. Can this data be used to align the QFormer so that task-related textual information (textual templates) is aligned properly to the subsequent module instead of adapting the LLM itself? Intuitively, you just need to align the qformer output with the specific language expected for input/output, and the data used to adapt the LLM itself could be used for this. I hope this is clearer. It was not an open criticism, just asking for clarification to try to understand this point better.
> >
> >
> > Regarding the question of: "I am a bit unsure about the impact of LoRA. A task prompt has much lower capacity to adapt to any downstream task compared to adding LoRA (especially if LoRA adapters are added to every 1x1 layer). So I'm wondering if part of the performance is due to LoRA vs no LoRA."
> >
> > I actually meant 1 - the instruction template works as a task prompt (in this case it is manually crafted instead of backpropagated as in prompt learning), but maybe I should have used "instruction template" as a name to make sure to avoid confusion.
> > InstructBLIP's task template has a much weaker modeling capacity than adding LoRA adapters to the LLM. It is thus natural to me that adding LoRA adapters result in numerically better results for the specific tasks/datasets (if data is adequate), as demonstrated in the table. I didn't mean to criticize using LoRA vs. another adaptation method, that's not the point. The question is, swapping a textual template for a LoRA does not seem like an apples-to-apples trade, as LoRA adds much higher flexibility and thus will work better in practice if fed decent data. We see a performance difference in the Tables. Is this because of the use of LoRA instead of textual templates?
> >
> > Both questions are related: I am not sure if LoRA are necessary and instead, the qformer could be tweaked by using the training data to align task templates. Adding LoRA adds much more capacity to mBLIP, and thus might leads to high performance not because of something methodological but because there's more modeling capacity thrown in. That is, it seems obvious that a task textual template will do worse than LoRA given the right data.

---

> > > ### Author Response · Authors · 2023-11-21
> > >
> > > Many thanks for your clarifications! We now much better understand your questions.
> > >
> > > **Could we fine-tune the Q-former with our multilingual prompts?**
> > >
> > > This would likely not work:
> > >
> > > BLIP2 initializes the Q-former with BERT and pre-trains the Q-former (without LLM at first) on various image-text objectives. InstructBLIP relies on the fact that the Q-former “understands” English and jointly feeds the prompt to both the LLM and the Q-former such that the latter produces prompt-contextualized tokens.
> > >
> > > BERT, however, only understands English and does not work well for other languages for two reasons:
> > >
> > > 1. The tokenizer does not work appropriately for non-Latin script languages
> > > 2. BERT has only seen English in language model pretraining
> > >
> > > Hence, the amount of data we use for languages other than English (e.g. Chinese, German, etc.) does not suffice to adapt BERT to those languages. Appropriate target-language adaption would then require both (1) extending the BERT tokenizer and (2) orders of magnitude more tokens in the target languages for language modeling pre-training.
> > >
> > > Consequently, if we wanted to use InstructBLIP’s approach, we would have to do the following:
> > > 1. Initialize a Q-former with a multilingual language model like XLM-R
> > > 2. Perform the very expensive, large-scale pre-training as in BLIP2 (6 days on 16 A100)
> > >
> > > We therefore believe that we cannot fine-tune the Q-former with our multilingual prompts.
> > >
> > > **Has LoRA more capacity than InstructBLIP’s task prompts?**
> > >
> > > For the above reasons, task prompts are not a viable option in the multilingual setting.
> > >
> > > We nevertheless agree that LoRA is expected to perform on par or better than InstructBLIP’s task prompts in a fair comparison. Ultimately, these approaches are not mutually exclusive.  Practitioners could jointly train a model (at least in English) on both approaches to likely boost performance further.

---

### Author Response · Authors · 2023-11-15
**General Response**

We thank all reviewers for their valuable time and feedback. We are happy to see appreciation for multilingual Vision-LLMs as a relevant topic [R1, R4], our proposed approach that it is computationally efficient yet yields performant multilingual Vision-LLMs [R2, R4], and a useful dataset for training multilingual Vision-LLMs for future work [R3].

R1 and R3 raise concerns about the novelty of our work. We concede that, on the modeling side, we primarily follow prior work; we highlight the novelties on the data/training side:

* We introduce the image-text matching task to Vision-LLM training and show that it can help reduce object hallucinations (Table 4).

* We propose to use back-translation consistency to automatically validate short VQA answer translations (PaLI, in contrast, only translates questions and never answers), which greatly helps with language hallucinations in VQA (see footnote 11).

* In addition, while translating training data is indeed an established method, prior work for multilingual vision-language models relied on commercial translation APIs, which can be prohibitively expensive at our data scale. We show that recent public translation models can be relied upon as robust (and free) alternatives.

Further, we would like to emphasize the practical impact of our work (an aspect that is sometimes neglected/underappreciated in comparison with technical/modeling novelty): to the best of our knowledge, mBLIP is the first massively multilingual instruction-based Vision-LLM that has been obtained with (i) only few-days of training effort on (ii) consumer-grade hardware and (iii) has publicly-released weights (i.e., the models will be publicly available). As such, it demonstrates that developing/training as well as using performant multilingual Vision-LLMs need not be the exclusive privilege of the large (industry) research groups with massive computational resources.

---

### Meta-Review · Area_Chair_ipow · 2023-12-05

**Metareview:**

The paper introduces an approach to developing multilingual vision-language models. The reviewers commend the cost-effective and technically sound methodology and the potential utility of the proposed dataset for future research. However, concerns about the paper's technical contribution, the novelty of the approach, and the representation of the multilingual dataset are significant. Additionally, there are issues with clarity and the need for more comprehensive comparisons with existing models. These concerns place the paper at a borderline level. The AC checked all the reviews and discussions, and believe the major concerns raised by the reviewers are valid. Thus, the paper is rejected.

**Justification For Why Not Higher Score:**

Concerns about the paper's technical contribution, the novelty of the approach, and the representation of the multilingual dataset are significant. Additionally, there are issues with clarity and the need for more comprehensive comparisons with existing models.

**Justification For Why Not Lower Score:**

N/A

---

### Decision · Program_Chairs · 2024-01-16

Reject